# Dataglove for Sign Language Recognition of People with Hearing and Speech Impairment via Wearable Inertial Sensors

**DOI:** 10.3390/s23156693

**Published:** 2023-07-26

**Authors:** Ang Ji, Yongzhen Wang, Xin Miao, Tianqi Fan, Bo Ru, Long Liu, Ruicheng Nie, Sen Qiu

**Affiliations:** 1Asset Management Department, Ketai Lexun (Beijing) Communication Equipment Co., Ltd., Beijing 101111, China; cheetah_ja@163.com; 2Scientific and Technological Innovation Center, Beijing 100012, China; wangyongzhen0204@126.com; 3Key Laboratory of Intelligent Control and Optimization for Industrial Equipment of Ministry of Education, Dalian University of Technology, Dalian 116024, China; mx1943563308@163.com (X.M.); ftq@mail.dlut.edu.cn (T.F.); boru@mail.dlut.edu.cn (B.R.); edaworld@163.com (L.L.); ruichengnie@mail.dlut.edu.cn (R.N.)

**Keywords:** multi-sensor information fusion, sign language recognition, wearable device, machine learning, deep learning

## Abstract

Finding ways to enable seamless communication between deaf and able-bodied individuals has been a challenging and pressing issue. This paper proposes a solution to this problem by designing a low-cost data glove that utilizes multiple inertial sensors with the purpose of achieving efficient and accurate sign language recognition. In this study, four machine learning models—decision tree (DT), support vector machine (SVM), K-nearest neighbor method (KNN), and random forest (RF)—were employed to recognize 20 different types of dynamic sign language data used by deaf individuals. Additionally, a proposed attention-based mechanism of long and short-term memory neural networks (Attention-BiLSTM) was utilized in the process. Furthermore, this study verifies the impact of the number and position of data glove nodes on the accuracy of recognizing complex dynamic sign language. Finally, the proposed method is compared with existing state-of-the-art algorithms using nine public datasets. The results indicate that both the Attention-BiLSTM and RF algorithms have the highest performance in recognizing the twenty dynamic sign language gestures, with an accuracy of 98.85% and 97.58%, respectively. This provides evidence for the feasibility of our proposed data glove and recognition methods. This study may serve as a valuable reference for the development of wearable sign language recognition devices and promote easier communication between deaf and able-bodied individuals.

## 1. Introduction

This article is aimed at individuals who experience hearing or speech impairments. Global statistics reveal that approximately 466 million individuals suffer from hearing loss, with 34 million being children. It is projected that by the end of 2050, around 900 million people will experience hearing loss [1]. Sign language serves as a critical form of communication for those who are deaf or mute, allowing them to express themselves and obtain important information. Sign language is primarily composed of finger language and gestures. Finger language pertains to the use of fingers to form letters, while gestures make use of hand movements, facial expressions, and body language to convey meaning [2]. Unfortunately, with the exception of those working in related fields, most individuals are unable to comprehend or communicate through sign language. This creates a significant communication barrier between individuals who are deaf or mute and those who are able-bodied. Deaf-mute sign language interpreters play a vital role in bridging the communication gap between individuals who are deaf or mute and those who are able-bodied. However, due to the shortage of deaf-mute sign language interpreters, their employment rates are high, and it is difficult to satisfy current societal needs. Consequently, the development of a barrier-free communication platform between individuals who are deaf or mute is critical, be it through communication with fellow deaf-mute individuals or with those who are able-bodied. Therefore, the aim of this paper is to propose a sign language recognition method for deaf people that combines multiple inertial sensors and advanced recognition algorithms to achieve more accurate sign language recognition results.

In recent years, significant progress has been made in studies on sign language or gesture recognition. Currently, mainstream sign language recognition methods are generally based on vision or sensor technology to capture relevant information from hand movements. This information is then combined with either machine learning [3] or deep learning [4] models to determine the corresponding meaning of sign language. One of the most commonly used methods for sign language recognition is based on computer vision. This method primarily uses a camera as its main tool to capture hand information, which is then analyzed to determine the meaning of the sign. This can be achieved through the use of a single camera [5,6] or multiple cameras [7]. In addition to traditional cameras, there are specialized cameras with specific functions that can be used for sign language recognition. For example, the Kinect camera [8,9,10,11] is capable of collecting depth information along with image data. This enables the capture of additional data, such as the subject’s depth information and skeleton data. With the help of its powerful software resources, it is possible to directly obtain position information of the hand in space, the direction angle of the hand, and other relevant information that can aid in sign language recognition. While using cameras as the core of sign language recognition can simplify the operation and provide a more comfortable experience for users by reducing the need for additional sensors, vision-based sign language recognition systems still have several drawbacks. Some of the main issues with these systems include changes in lighting conditions, variations in the environment, high computing costs, and limited portability [12].

On the other hand, sensor-based sign language recognition systems can compensate for the limitations of vision-based systems. Currently, sensors used for gesture or sign language recognition can be divided into two types: those that collect biological signals and those that collect abiotic signals. The former generally collects the EMG (Electromyographic) signal or skin impedance signal of the user’s hand or arm. There have been several notable studies on using EMG and skin impedance signals for sign language recognition. For example, Xilin Liu et al. [13] proposed a gesture recognition system based on four differential EMG channels. This system was capable of recognizing ten gestures operated on a touchpad with a mass of only 15.2 g and an endurance of up to 40 h, making it a noteworthy reference in terms of quality and power consumption. Similarly, Dai Jiang et al. [14] developed a 16-electrode electrical impedance tomography system that could recognize eight static gestures by means of electrodes placed at the wrist and above the forearm. For sensors that collect biological signals, precise sensor placement is essential and may require professional assistance for installation guidance.

In contrast, sensors that collect abiotic signals have lower placement requirements and can be easily incorporated into wearable devices for sign language recognition. Li et al. [15] utilized accelerometers and gyroscopes integrated into smartwatches to gather muscle activity data for identifying subtle finger gestures. Similarly, Wang et al. [16] employed a single inertial measurement unit (IMU) placed at the wrist in conjunction with machine learning classification methods to explore combined gesture recognition. Tai et al. [17] employed smartphones in combination with long- and short-term memory neural networks (LSTM) to explore continuous recognition of six types of gestures. These sensors are not restricted by the environment and are therefore more appropriate for sign language recognition among the deaf and mute. 

However, the recognition of complex and fine dynamic sign language movements using only one sensor is challenging, which makes achieving outstanding recognition performance difficult. In light of these limitations, Dong et al. [18] developed a low-cost and straightforward data glove featuring integrated IMU and bending sensors capable of capturing finger motion and bending. Meanwhile, Calado et al. [19] designed a wearable electronic system that includes a pair of gloves equipped with ten flexible sensors and six IMUs to measure the motion information of the hand, arm, and forearm. Pan et al. [20] conducted studies on the significance of different sensor combinations for accurate sign language classification. The team designed a capacitive pressure glove that utilized 16 capacitive sensors to capture hand gestures, taking into consideration data throughput in wireless transmission. Their work has inspired our system’s framework design.

Studies conducted on data gloves have provided us with extensive inspiration. Building upon these insights, our team has designed a multi-IMU data glove capable of capturing posture, motion, and position information of the hand, fingers included, all while maintaining data consistency. During hardware design, we ensured that the data glove is small, lightweight, and comfortable to wear. In particular, each data glove weighs only 37.8 g and is powered by a 500 mAh lithium battery, which guarantees up to two hours of use. The design of a data glove based on a single type of sensor significantly reduces the computation required during post-sequence data processing. However, for accurate sign language recognition involving high similarity, multi-IMU designs are necessary. Our system can thus function as an independent sign language recognition system that does not rely on external sensors. The primary contributions of our studies are as follows:Our team designed a data glove based on multi-IMU sensors capable of capturing complex movements in sign language. A single data glove integrates 16 IMUs, allowing for the accurate capture of intricate sign language gestures.Our team evaluated the impact of the number and location of motion analysis nodes on recognition performance and successfully reduced the cost and computational complexity of the equipment, making it more acceptable to the public.We propose a dynamic sign language recognition solution that uses four machine learning methods: decision tree (DT), support vector machine (SVM), k nearest neighbor (KNN), random forest (RF), and a bidirectional short-term memory neural network deep learning model based on attention mechanism (Attention-BiLSTM) to recognize 20 kinds of dynamic sign language. The proposed method is compared with the existing advanced methods on the common data set.

The structure of this study paper is as follows: Section 2 briefly presents the system architecture design of the proposed sign language recognition system as well as the hardware design of the multi-IMU data glove. Section 3 discusses the data collection process, data preprocessing methods, and feature extraction techniques utilized in the sign language recognition experiments. Section 4 presents the experimental results of traditional machine learning methods and deep learning algorithms and compares their performance from multiple perspectives. Section 5 summarizes the key findings of our studies and discusses future directions for sign language recognition technology.

## 2. System Architecture and Platform Design

The objective of this study is to create an affordable and portable set of wearable smart gloves for the recognition of Chinese sign language. The gloves are designed to capture various features of hand and finger gestures, such as direction, position, and amplitude, which can be used in sign language recognition studies. 

The hardware design of the smart gloves comprises two main components:Motion Acquisition Node (MAN): This component includes multiple sensors, such as accelerometers and gyroscopes, that capture motion data from the wearer’s hand and fingers.Data Aggregation Node (DAN): This component aggregates the motion data collected by the MAN and sends it to the recognition system for further analysis and processing.

Figure 1 depicts the overall hardware system framework for the smart glove. Each glove has 16 Motion Acquisition Nodes (MANs), which are equipped with an MPU9250 9-axis sensor with specific parameters, as shown in Table 1. The Data Aggregation Node (DAN) receives and aggregates the motion data collected by MANs, adds header and trailer information to facilitate subsequent data uploads, and transmits the data wirelessly using an onboard ESP8266 WIFI module from Espressif. This allows for the wireless upload, storage, and processing of motion data by the acquisition software. Overall, the hardware design of the smart glove enables the capture of detailed motion data necessary for recognizing Chinese sign language, and the wireless capabilities of the DAN allow for convenient data transmission and processing. The processor used for the smart glove is the STM32407VGT6 chip, which is based on the Cortex-M4 framework. The system is powered by a centralized 3.3 V lithium battery.

Figure 2 provides an actual image of the designed data glove. Each finger of the glove contains three IMUs (one on the thumb, index finger, middle finger, ring finger, and little finger), which measure the movement information of the fingers. The IMU nodes are connected using soft wires to ensure flexibility in joint movement. 

This design allows for the capture of detailed and accurate hand and finger movements necessary for recognizing Chinese sign language gestures. To measure the motion information of the hand as a whole, an IMU is strategically placed on the back of the hand. The mapping of each sensor to its specific location on the hand and the physical diagram of the wearable body are illustrated in Figure 3. 

The hardware transmission protocol for the entire system involves 16 MANs, each containing 18 bytes of data, resulting in a total of 288 bytes. To account for packet verification, a header is added, bringing the total number of bytes to 289. The MANs are connected via an SPI bus, and the DAN receives the data by selecting the appropriate address chip on the bus. The DAN then fills a data buffer with the relevant bytes from each MAN in sequence. Figure 4 depicts the structure for data transmission. Once all MANs are received, the DAN forwards the data to the host computer via the router. The network communication protocol employed is TCP, which guarantees that data will not be lost or become out of order during wireless transmission. The system’s sampling frequency is set at 100 Hz, ensuring the collection of accurate and reliable data. The software platform is composed of two main components: the control and analysis parts. The control part of the platform sends acquisition instructions to the glove through the TCP network protocol while monitoring its working status. This enables control over the data acquisition process and allows for seamless and efficient management of the collected data. The analysis part of the platform processes the original inertia data collected by the dataglove, reconstructs the hand posture, and identifies sign language gestures. After conducting several experiments, the platform has been optimized to meet the specific requirements of the task at hand, making it suitable for further studies and development.

## 3. Sign Language Recognition

### 3.1. Initialize Attitude and Magnetometer Calibration

The sensor used in the inertial glove is a cost-effective MPU9250, which has significant gyroscopic drift during hand attitude solving. This can result in inaccurate posture reconstruction and often requires the use of magnetometer data for fusion compensation. However, the magnetometer is easily influenced by various factors in its surroundings, leading to large measurement errors. Therefore, correction techniques are needed to improve the accuracy of the measurements. This paper uses the eight-character calibration method to calibrate the magnetometer. After the data glove is activated, it rotates 3–5 times in a figure-eight motion through the air. As shown in Figure 5a, the output data of the magnetometer is affected by environmental interference and self-error, resulting in an ellipsoid-shaped distribution with the center not located at the coordinate origin. This method helps to correct these errors and improve the accuracy of the magnetometer measurements. To calibrate the magnetometer, this paper employs the least squares method to solve the ellipsoid fitting problem. Figure 5b shows the output of the calibrated magnetometer, where the measured values fit the sphere located at the origin. This result indicates that the calibration process has effectively addressed environmental interference and self-error issues, resulting in accurate and reliable measurements.

### 3.2. Hand Posture Calculation

The participants wear the inertial gloves and wait for the initialization process to complete, putting them in standby mode. Meanwhile, the PC software connects to the router’s local area network to establish communication with the inertial gloves. Once connected, an acquisition instruction is inputted into the PC and transmitted to both gloves through a TCP point-to-point connection to start collecting inertia data from the subject’s hand. This procedure ensures seamless communication between the gloves and the software, facilitating efficient data collection and analysis. On the PC side, the software uploads and stores the inertia data collected from both gloves upon receiving the start instruction. Figure 6 provides an example of the original data collected from a single MAN on one of the gloves. This data serves as the basis for subsequent processing and analysis to reconstruct posture and recognize sign language gestures. The output data of the inertial gloves is based on their respective sensor coordinates, while the sign language gestures analyzed in this paper are relative to spatial positions. To achieve this, the study involves a variety of commonly used coordinate systems and requires conversion between them. Figure 7 illustrates the three coordinate systems used in the system, which are as follows:

Sensor coordinate system (SCS, O-XsYsZs), which is based on the gyroscope instrument in the MPU9250 data manual. When the inertial gloves are worn, the SCS coincides with the carrier coordinate system, thus reducing errors in attitude calculations.Body coordinate system (BCS, O-XbYbZb), which takes the centroid of the carrier as the origin. In this paper, the three axes of the BCS point in the front, right, and bottom directions.Geographic coordinate system (GCS, O-XgYgZg), which is based on the position of the carrier. In this paper, the orientations of the three axes of the GCS are set to north, east, and geocentric, respectively.

It should be noted that in this study, the term “north” specifically refers to the magnetic field’s north pole. Moreover, the attitude calculation of the hand takes into consideration the issues of Euler angle universal lock and the high computational complexity of the rotation matrix. As a result, the study utilizes quaternions to describe the three-dimensional attitude, as depicted in Equation (1).
(1)$$\vec{q}=q_{0}+q_{1}\vec{i}+q_{2}\vec{j}+q_{3}\vec{k}$$
where $q_{0}$ is the scalar part of $\vec{q}$, $q_{1}$, $q_{2}$ and $q_{3}$ is the vector part and $i^{2}=j^{2}=k^{2}=-1$.

The relationship between the attitude angle and the quaternion can be expressed as shown in (2).
(2)$$\left\{ \begin{array}{l} \mathrm{roll}=\arctan\left(\frac{2(q_{2}q_{3}+q_{0}q_{1})}{q_{0}^{2}-q_{1}^{2}-q_{2}^{2}+q_{3}^{2}}\right)\\ \mathrm{pitch}=\arcsin(-2(q_{1}q_{3}-q_{0}q_{2}))\\ \mathrm{yaw}=\arctan\left(\frac{2(q_{1}q_{2}+q_{0}q_{3})}{q_{0}^{2}+q_{1}^{2}-q_{2}^{2}-q_{3}^{2}}\right) \end{array}\right.$$
where the term “roll” refers to the rolling angle, which is the angle of rotation of the hand relative to the Yg axis of the Geographic Coordinate System. Similarly, “pitch” is the pitch angle, which represents the angle of rotation of the hand around the Xg axis relative to the Geographic Coordinate System, and “yaw” denotes the heading angle, which is the angle of rotation of the hand around the Zg axis relative to the Geographic Coordinate System.

### 3.3. Construction of Sign Language Dataset

To evaluate the effectiveness and versatility of the system, the paper designed 20 sign languages based on commonly used sign language in daily communication among Chinese deaf-mute individuals, as depicted in Figure 8. Table 2 provides a description of the twenty sign language messages used in the study. 

The authors selected seven representative participants—six males and one female—to collect sign language data. The average height of the participants was 175 ± 15 cm, and the average weight was 60 ± 15 kg.

The first step of the data collection process involved turning on the glove switch to initialize the system. The PC software was then connected to begin collecting sign language data. Additionally, before starting the data collection process, participants were instructed to face north for a duration of 3–4 s to complete the initial attitude calibration. During the data collection process, the participants were prompted by the screen to complete each sign language action. After completing each action, the participants were instructed to let their arms sag naturally. Once a set of sign language data was collected, the original inertia data of the sign language was saved to a designated folder as sample data for later processing. After each set of data collection, there was a two-minute break before proceeding to the next set. Each subject completed a total of 10–20 sets of sign language data collection.

### 3.4. Sign Language Segmentation and Feature Extraction

After pre-processing, the 3D raw acceleration signals and 3D angular velocity signals obtained from all MANs are combined with the pose quaternions obtained from pose decomposition to construct a 160-dimensional data structure, as depicted in Figure 9. This data structure will then undergo feature extraction. 

For this study, sign language data was manually segmented by comparing it with high-frame-rate camera video. Since the data from each MAN is sampled synchronously, only the MANs at the back of the hand need to be segmented. Each sign language action was ensured to be completed within 4 s, and there was an interval of 2 to 3 s between sign languages. Each sign language was labeled as 1 to 20, respectively. Figure 10 shows the segmentation results of some of the sign languages after undergoing the smoothing process.

In this paper, feature extraction was carried out using the sliding window method. The sign language time series, as shown in Figure 10, was segmented into a time window sequence consisting of 400 sample points, which corresponds to 4 s of sign language data. The window was then moved forward by 20 sample points (or 0.2 s) at a time.

As new sign language information is received, the sliding window continues to move forward in the pre-processed data. The latest window data is then sent to the classifier for sign language recognition. 

Feature extraction is a crucial step in improving the accuracy of sign language recognition models. For this study, feature extraction was performed on each dimension of the 160-dimensional time series obtained through preprocessing. This resulted in a total of 1280-dimensional feature data. This high-dimensional feature data set can help capture detailed information about the sign language signals and improve the accuracy of the classification model. The features extraction process consists of eight common feature variables used in both time domain and frequency domain time series analysis, namely mean, standard deviation, skewness, kurtosis, crest factor, quartile distance, spectral kurtosis, and spectral frequency. Detailed descriptions and the main formulas for feature extraction are shown in Table 3. For instance, we took MAN 1 of the thumb and extracted features from its 3D acceleration signal, 3D gyroscope signal, and quaternion, as depicted in Figure 11. These features were combined to generate an 80-dimensional feature data set for MAN 1. The same process was repeated for the other 15 MANs. Next, label information for the gesture sequences described in the previous section was assigned to the feature variables obtained from each window. During the labeling process, if a feature variable from the current gesture phase does not contain any information from the next gesture phase, it is assigned a label of 1, and similarly for subsequent phases. However, if a feature data set contains any information about the next gesture phase, it is discarded to prevent any confusion during the classification process.

### 3.5. Sign Language Segmentation and Feature Extraction

The feature data extracted using the methods described above will ultimately be fed into the machine learning recognition model in the form of a vector. This study explores the following four algorithm models, which are commonly used in classification and recognition studies:Support Vector Machine (SVM): The SVM is a binary classification model in which an SVM model is designed between any two samples using the one-to-one method. When classifying unknown samples, the sample type is determined by the highest score;Decision Tree (DT): DT is a commonly used machine learning algorithm. In this paper, the C4.5 algorithm is utilized to construct the decision tree, and the optimal partition attribute is selected for each node;K-Nearest Neighbor (KNN): This is a commonly used supervised learning method in which the KNN algorithm is implemented with K set to 5. The weights of the proximity points are equal, and the Euclidean distance measure is utilized;Random Forest (RF): RF is a method of integrating multiple weak classifiers into a single strong classifier to classify the target. In this paper, we implement the typical RF algorithm of the bagging algorithm to solve the multi-classification problem, with the number of learners set to 100.

### 3.6. Deep Learning Recognition Algorithm Model

In addition to the four aforementioned algorithm models, this paper also explores the use of a bidirectional long-short-term memory (Bi-LSTM) based on LSTM. Bi-LSTM combines both the forward and reverse information of input sequences, resulting in better performance in sequence labeling tasks. The model framework of a single-layer Bi-LSTM is depicted in Figure 12. However, Bi-LSTM adopts the traditional encoding-decoding method, and the sequence samples are edited into fixed-length vectors regardless of their lengths prior to being inputted to the Bi-LSTM model. Given the differences in sequence lengths of various sign language samples in practical applications, there is a possibility that certain key factors may be overlooked during model training, possibly resulting in poor recognition performance of the model. To address this issue, an attention mechanism is introduced in this paper. By introducing the attention mechanism, we can break the problem of fixed vector length in the encoding process of Bi-LSTM and give the corresponding weights according to the characteristics of the sequence to show the key information more clearly, which can improve the model’s training efficiency and help the model make accurate recognition.

The structure of the attention mechanism is depicted in Figure 13, where $y_{t}^{}$ represents the hidden layer vector outputted by Bi-LSTM at each time step denoted as key, and $y_{n}$ represents the output at the last time step denoted as query. The computation process is as follows:(3)$$S_{t}=\alpha (y_{t},y_{n})$$
(4)$$a_{t}=\frac{\exp(S_{t})}{\sum_{t=1}^{n}\exp(S_{t})}$$
(5)$$c=\sum_{t=1}^{n}a_{t}y_{y}$$
where $S_{t}$ is the similarity score between $y_{t}$ and $y_{n}$ at each moment calculated by the learning function α, and then it is normalized by the softmax function to obtain the weight $a_{t}$ of $y_{t}$ at each moment, and finally the vector c is calculated by Equation (5).

In this paper, we have designed an Attention-BiLSTM model that comprises an input layer, two Bi-LSTM layers, two Dropout layers, an Attention layer, a fully connected layer, and an output layer. The sign language acceleration, gyroscope, and quaternion data collected are manually segmented and labeled. These labeled data are then provided to the input layer of the model. The Bi-LSTM layer performs the initial feature learning of the sign language information. Next, it is passed through the Dropout layer and inputted to the lower Bi-LSTM layer for the second stage of feature learning. Then, it is passed through the Dropout layer again and inputted into the Attention layer to calculate the similarity score. The output from the Attention layer is normalized by the softmax layer to calculate the weight information at each moment. This result is then sent to the fully connected layer, where the output layer generates the sign language recognition result. We chose to develop the model based on Bi-LSTM because the inertial sensor signal of sign language is time-dependent, and each sign language gesture contains a lot of information in the time domain. The Bi-LSTM layer is well-suited to capture contextual information in the sequence. Also, since we had a relatively small number of sign language data samples, Bi-LSTM was a better choice, as it is particularly useful for small sample datasets. The Dropout layer is utilized to address the problem of overfitting in the model. In conclusion, the use of Attention-BiLSTM can improve training efficiency and recognition accuracy while performing bidirectional semantic learning of sign language time series.

In this paper, the aforementioned classification algorithms were employed to accomplish the recognition and prediction of 20 sign languages. All machine learning algorithms were run in the Python SK-learn library, while the Attention-BiLSTM model was executed in MATLAB 2022a on a system with an AMD RX-5800H CPU running at 3.20 GHz and a RX3060 GPU, with 8.00 GB of RAM, and operating on Windows 11.

## 4. Results and Discussion

In this section, we perform empirical experiments on 20 sign languages that were collected with data gloves in order to validate the performance of the data gloves as well as the feasibility of the proposed sign language recognition method.

In order to decrease the computational demands of sign language recognition and reduce the power consumption and cost of the system, we evaluated the impact of the number and placement of the motion analysis nodes (MANs) on the recognition performance. We designed eight different combinations, as outlined in Table 4, and used the model’s recognition accuracy as the measurement index. Among the 8 combinations, we used all of MAN 1, mainly because the motion of the whole hand is essential in dynamic sign language, while the other MANs are arranged and combined according to the different joint positions of each finger by arranging and combining different nodes to achieve recognition accuracy without reducing power consumption or computation. Based on the results shown in Figure 14, it can be observed that there is a drastic reduction in recognition accuracy when using only the position data from MAN 1, while adding more than 6 MANs do not lead to any significant increase in recognition accuracy. This can be attributed to the fact that dynamic sign language gestures involve more spatial motion of the hand and less complex motion of the fingers as compared to static sign language. The relatively higher accuracy of combination 4 with the same 6 MANs may be due to the linkage structure of the finger, where the motion of the distal knuckle incorporates the motion characteristics of the proximal knuckle; for example, when the distal knuckle moves, the proximal knuckle follows, which also means that it has a faster acceleration and amplitude of motion. In subsequent experiments, we will use the MANs from combination 4 to further evaluate the performance of the model.

If the training set is too large or imbalanced, or if the model is overly complex, there is a risk of overfitting the recognition model. This means that the model performs well on the training set but poorly on the validation set. Conversely, if the size of the training set is too small, there is a risk of underfitting the model and missing important patterns and correlations in the data. In other words, if the model’s recognition performance is poor on both the training and validation sets, it may indicate a lack of balance in model complexity and dataset size. A high-quality recognition model should aim to find a balance between the two to ensure the model has real-world practicality beyond merely achieving good performance on the training set. Therefore, this study comprehensively considers the practical situation by utilizing a five-fold cross-validation approach. This method involves dividing the dataset into five subsets, using four subsets for training and one subset for validation, and repeating the process five times, ensuring each subset is validated once. Ultimately, the final prediction result is achieved by averaging the results of these five validations. The total number of 20 sign language samples involved in training and validation is 2219 groups, of which each sign language sample is around 110 groups to ensure a balanced sample. 90% of the samples are selected for five-fold cross-validation, and 10% are used for final result validation.

Accuracy is typically a crucial metric used to evaluate the performance of multi-classification models, as represented mathematically in Formula (6).
(6)$$\mathrm{Accuracy}=\frac{\mathrm{TN}+\mathrm{TP}}{\mathrm{TP}+\mathrm{FN}+\mathrm{FP}+\mathrm{TN}}$$

The parameters of the deep learning model were carefully configured in light of potential issues such as underfitting or overfitting resulting from small training sets or unbalanced samples. For the Attention-BiLSTM model, the learning rate was set to 0.001, and the batch size for training samples was 256. Additionally, the epoch was set to 100. It’s worth noting that TP refers to true positive, FP refers to false positive, TN represents true negative, and FN indicates false negative, all of which are necessary for assessing model performance. Figure 15 displays the training process, testing process, and loss function of the final model. It is evident that the loss function of both the training and testing processes ultimately converges, and the accuracy of the model achieves excellent results. In this paper, we utilized a machine learning model that employed a grid search algorithm to optimize its parameters. Ultimately, we selected a decision tree with information entropy as the feature criterion. Additionally, we set the parameter C for the SVM to 0.01 and chose the radial basis function as the kernel function. The K value for KNN was chosen as 5, and we used 100 evaluators in the RF model.

Figure 16 presents the average accuracy of the five models for 20 sign language recognition tasks. It is evident that both RF and Attention-BiLSTM achieve an accuracy of over 97%. While RF is a traditional machine learning method requiring manual feature extraction, the latter utilizes deep learning technology for implicit, self-completed feature extraction through the Attention-BiLSTM model. The average accuracy for RF and Attention-BiLSTM achieved in this study is 97.58% and 98.85%, respectively. The details of the five cross-validations are presented in Table 5, indicating that the accuracy of all five models used in this study is stable following the five-fold cross-validation.

While accuracy is an important performance metric, it only offers a general overview of a model’s classification ability and fails to reflect how well it performs on specific classes or generalizes to new data. To address this issue, we introduced three additional evaluation indicators—accuracy, recall, and F_1_-score—to measure the model’s generalization performance more comprehensively. Accuracy is limited to positive samples that are predicted correctly, specifically showing how many true positives exist within the positive predictions. The accuracy function is shown in Equation (7).
(7)$$\mathrm{Precision}=\frac{1}{n}\sum_{i=1}^{n}\frac{{\mathrm{TP}}_{i}}{{\mathrm{TP}}_{i}+{\mathrm{FP}}_{i}}$$

The Recall rate indicates the percentage of actual positive samples correctly identified by the classifier, as demonstrated in Equation (8).
(8)$$\mathrm{Recall}=\frac{1}{n}\sum_{i=1}^{n}\frac{{\mathrm{TP}}_{i}}{{\mathrm{TP}}_{i}+{\mathrm{FN}}_{i}}$$

The F_1_-score is a holistic performance metric that combines the precision and recall of the model, yielding a balanced evaluation. Equation (9) demonstrates the calculation of the F1-score.
(9)$$F_{1}\text{-score}=\frac{2\cdot \mathrm{Precision}\cdot \mathrm{Recall}}{\mathrm{Precision}+\mathrm{Recall}}$$

After performing five-fold cross-validation on the five model methods, we selected the model with the highest accuracy to evaluate its generalization performance. We fed the prepared dataset containing samples that were not involved in training and verification into this model and obtained the final evaluation results, as presented in Figure 17. It is evident from the figure that Attention-BiLSTM exhibits the best generalization performance, followed by RF. A detailed breakdown of the five evaluation indicators for each model is recorded in Table 6.

Our study compared traditional machine-based algorithms with deep learning algorithms, building on the studies of Matej Králik et al. [21]. We utilized two sign language datasets (waveglove-single and waveglove-multi) that they had gathered and shared publicly, as well as six public datasets for human activity recognition that were standardized in [22] and uWave [23]. Table 7 displays a publicly available gesture dataset alongside its associated accuracy evaluation metrics. The results demonstrate that our feature extraction method combined with RF achieves recognition accuracy far higher than that of the deep learning model on several public datasets. However, our Attention-BiLSTM model still achieves the highest accuracy among all six public datasets, indicating the superiority of our proposed method. Nonetheless, there is still scope for improvement.

Figure 18 and Figure 19 below display the confusion matrix diagrams of the two models with higher recognition performance, namely Random Forest and Attention-BiLSTM. Figure 18 depicts the recognition results of the RF model, while Figure 19 represents those of the Attention-BiLSTM model. The diagonal elements in the confusion matrices indicate the proportion of model predictions that match the true sample labels. Upon examination of the figures, we observe that the Random Forest model achieves an accuracy of at least 96% for most of the sign languages. Only five sign languages have recognition accuracy lower than 96%. In contrast, the Attention-BiLSTM model recognizes most of the sign languages with high accuracy, except for “they,” “goodbye,” “joke,” and “stand up.” The recognition accuracy for the remaining 16 sign language categories reaches 100%. Based on our analysis, the reason for the low accuracy of the aforementioned sign languages is their relatively simple movements and lack of finger joint movements. This makes it difficult to extract useful information about the fingers during feature extraction and sequence learning due to the small amplitude of finger movements and short execution time.

Even though the Attention-BiLSTM model outperforms traditional machine learning for recognition accuracy, this doesn’t mean that deep learning is always superior. As [26,27] notes, traditional models rely more on data features and are generally simpler. Therefore, deep learning may not always be necessary or practical, as Random Forest provides adequate results in some cases. Ultimately, the choice of approach depends on the specifics of the problem, such as data availability, computational resources, and time constraints. In this study, we extracted only eight feature variables from the time series for manual feature extraction, which yielded promising results in the RF model. However, this approach also allows for a more comprehensive understanding of the data and the underlying algorithm when compared to the black box structure of deep learning models. In practical engineering fields, traditional machine learning methods tend to have significantly lower computational costs than deep learning methods. Wearable devices for sign language recognition must consider factors such as portability, power consumption, cost, and comfort [28]. Given these considerations, it can be challenging to install the necessary computational units for deep learning, preventing the realization of performance benefits associated with deep learning models. Traditional machine learning models offer the benefits of fast training and simple deployment, allowing for faster update iterations in hardware products. As engineering costs are mainly focused on data processing and feature optimization in the early stages of the model, different model approaches can be tested in a short period of time. Deep learning is not yet capable of achieving these aspects at this stage.

## 5. Conclusions

In this paper, we design a multi-IMU-based data glove that captures hand gestures and finger motions for the recognition of 20 instances of deaf sign language. To evaluate the data glove system’s recognition performance, we used both traditional machine learning and deep learning approaches. Our traditional machine learning experiments involved four models: DT, SVM, KNN, and RF. For our deep learning approach, we used the Attention-BiLSTM model. The two methods with the highest classification accuracy were RF (97.58%) and Attention-BiLSTM (98.85%), demonstrating the feasibility of our proposed data glove for sign language recognition. Finally, we discuss the potential application of our sign language recognition algorithm to wearable devices.

In our future work, we aim to expand our sign language data sets and improve the generalization and accuracy of our models. We will address the issue of automatic division of training data and explore the benefits of a two-handed recognition model. Our ultimate goal is to utilize data glove technology and machine learning to enhance communication and accessibility for deaf individuals. In addition, we will focus on optimizing the data glove by developing a flexible circuit board, reducing node size, enhancing human-computer interaction functions, and designing a more wearable device. Our ongoing efforts aim to improve the user experience and ultimately help break down barriers for individuals with hearing impairments.

## Figures and Tables

**Figure 1 sensors-23-06693-f001:**
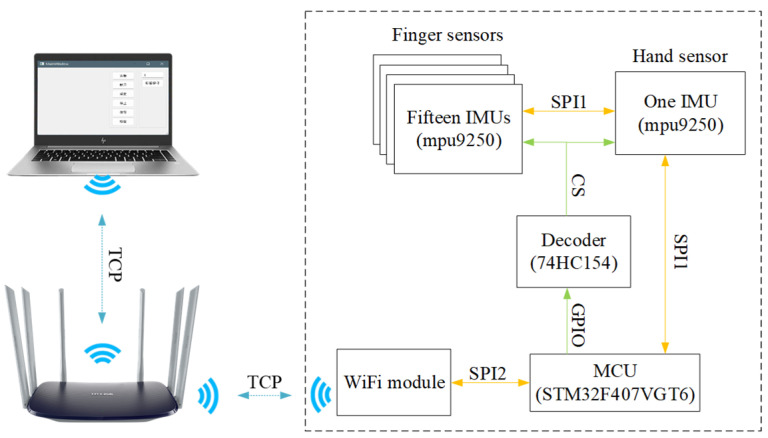
System architecture of gloves.

**Figure 2 sensors-23-06693-f002:**
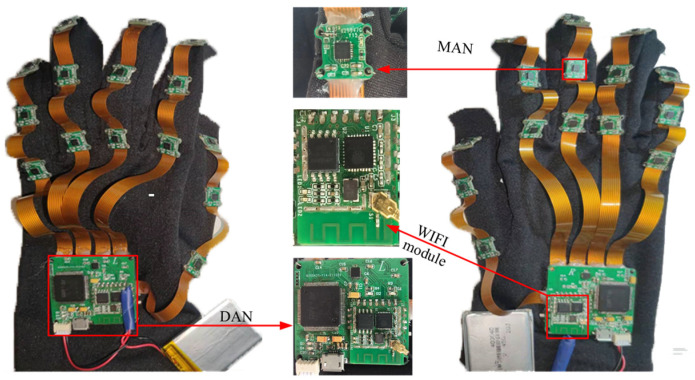
Physical picture of data glove.

**Figure 3 sensors-23-06693-f003:**
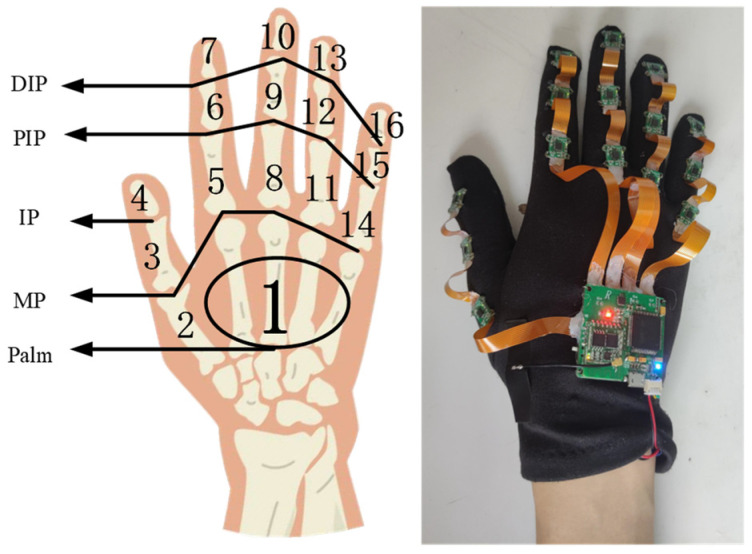
Mapping relationship between inertia node and hand (right hand). The numbers in the left figure represent the sensor node serial numbers.

**Figure 4 sensors-23-06693-f004:**
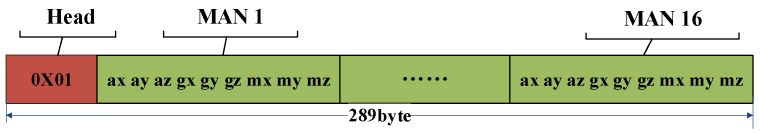
Data transmission structure.

**Figure 5 sensors-23-06693-f005:**
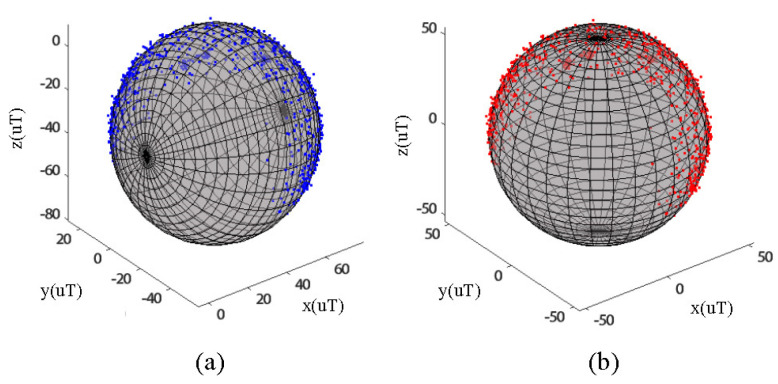
Magnetometer before correction (**a**), Magnetometer after correction (**b**).

**Figure 6 sensors-23-06693-f006:**
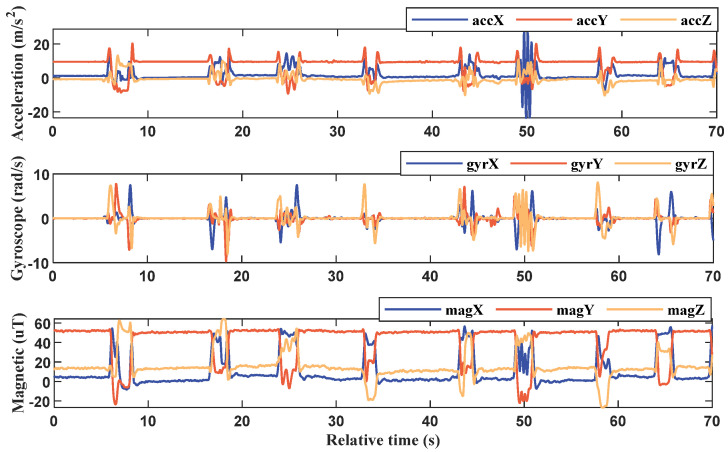
Data glove single MAN raw data.

**Figure 7 sensors-23-06693-f007:**
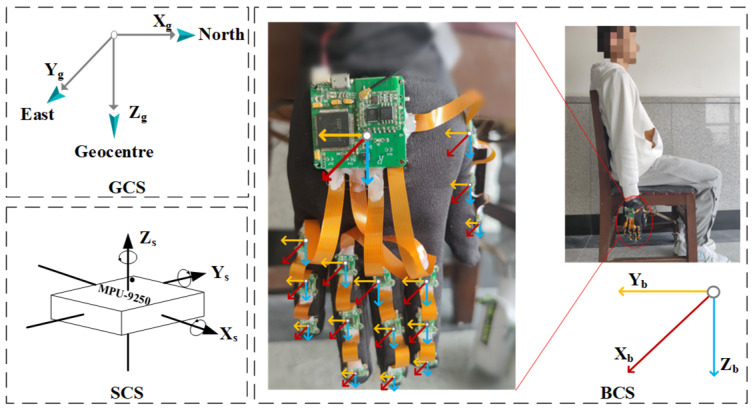
Definition of coordinate systems.

**Figure 8 sensors-23-06693-f008:**
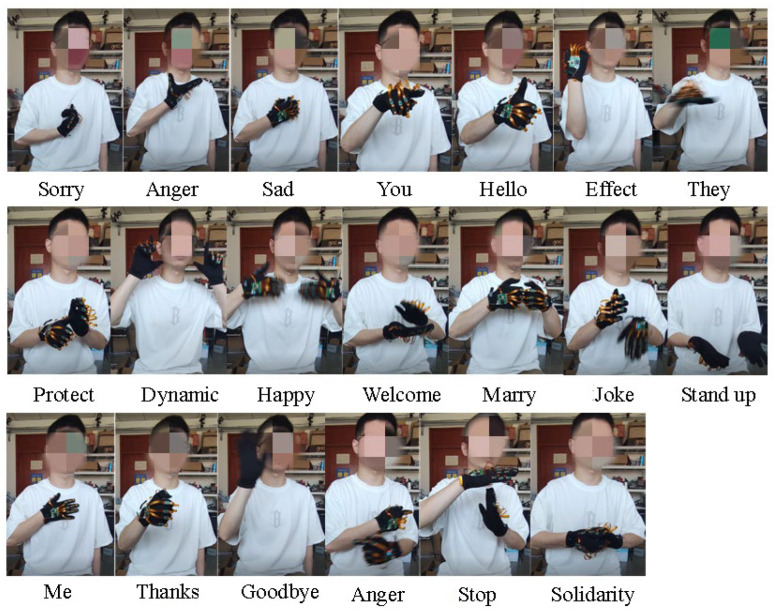
20 kinds of Chinese sign language actions.

**Figure 9 sensors-23-06693-f009:**
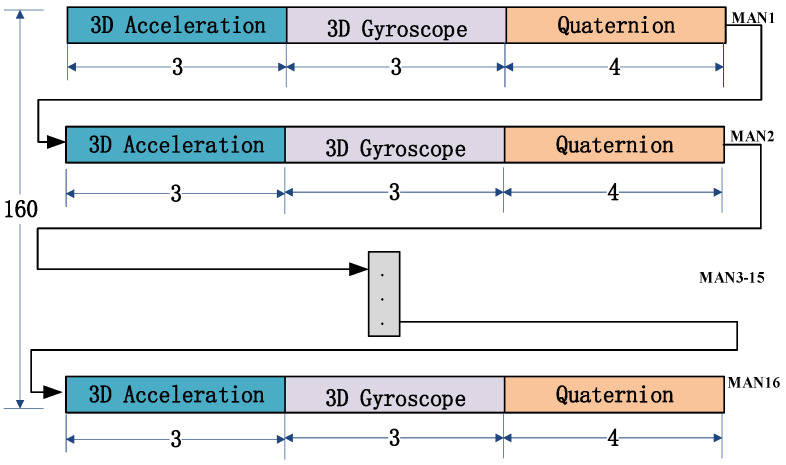
Preprocessing data structure.

**Figure 10 sensors-23-06693-f010:**
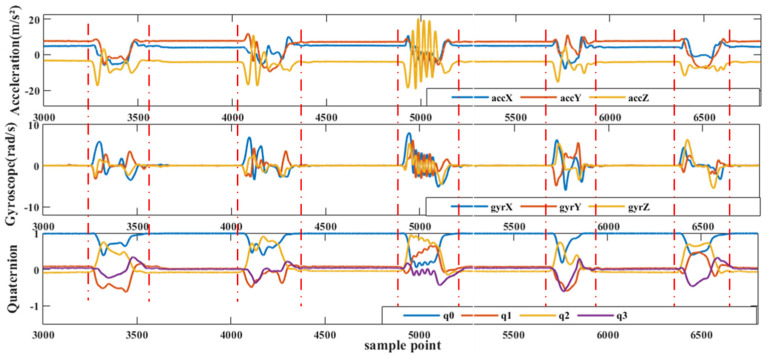
Sign language data segmentation.

**Figure 11 sensors-23-06693-f011:**
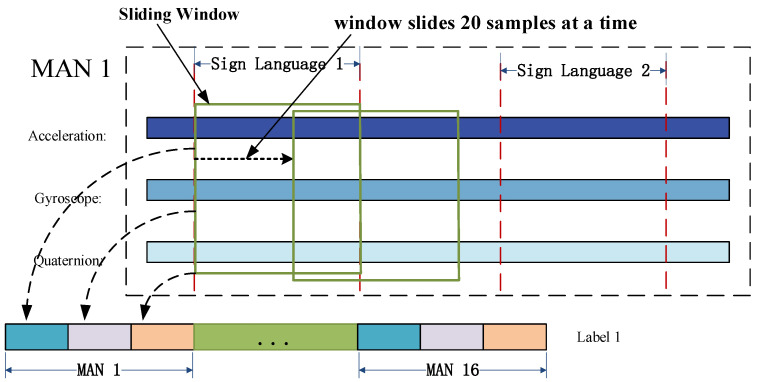
Features extraction process.

**Figure 12 sensors-23-06693-f012:**
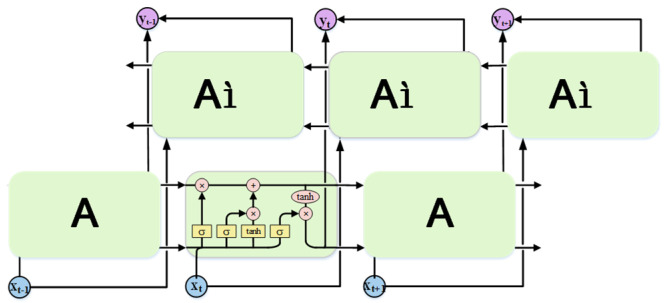
Bi-LSTM model framework.

**Figure 13 sensors-23-06693-f013:**
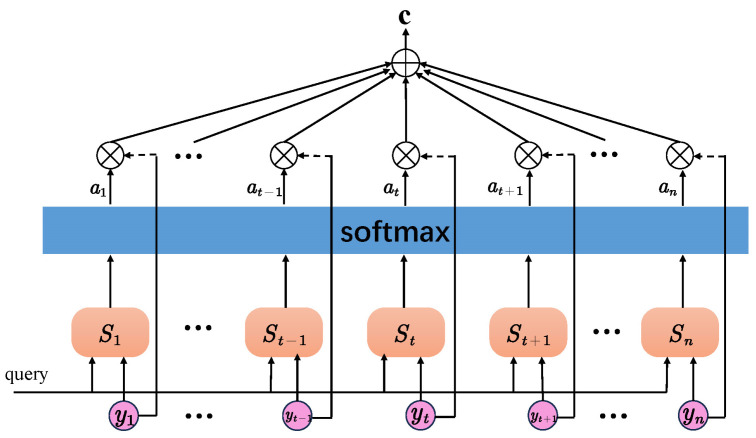
Structure of attentional mechanisms.

**Figure 14 sensors-23-06693-f014:**
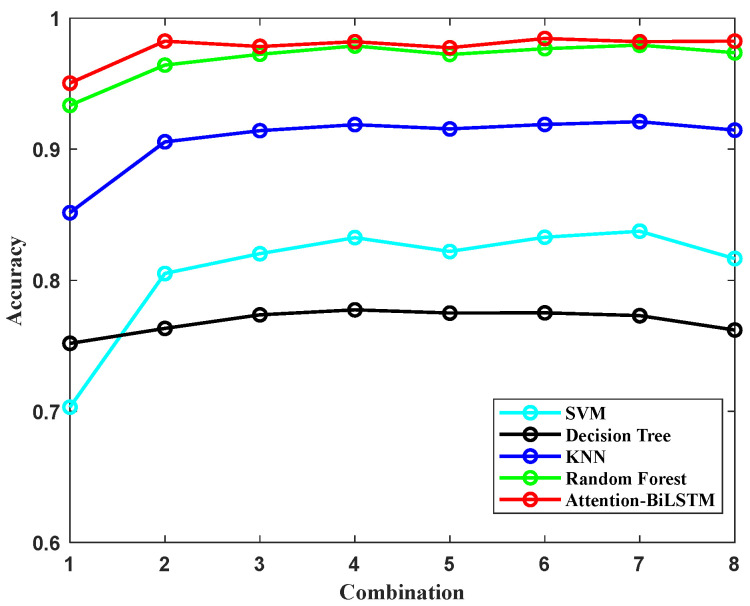
Average accuracy of underhand speech recognition for different combinations of MANs.

**Figure 15 sensors-23-06693-f015:**
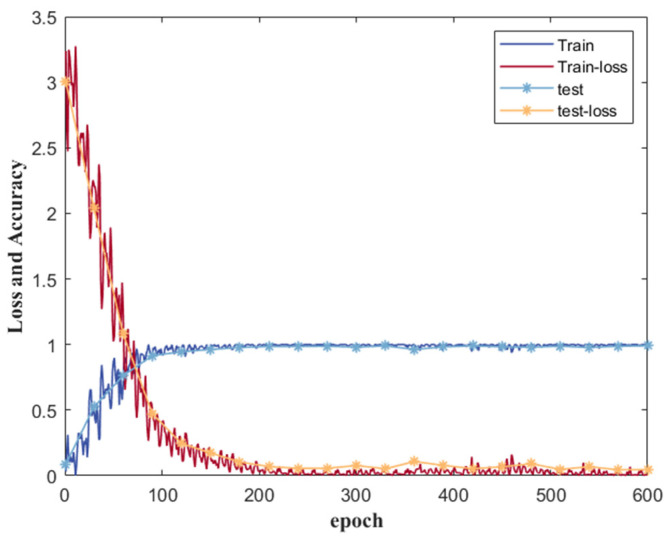
Training effect of Attention-BiLSTM Model training and test sets.

**Figure 16 sensors-23-06693-f016:**
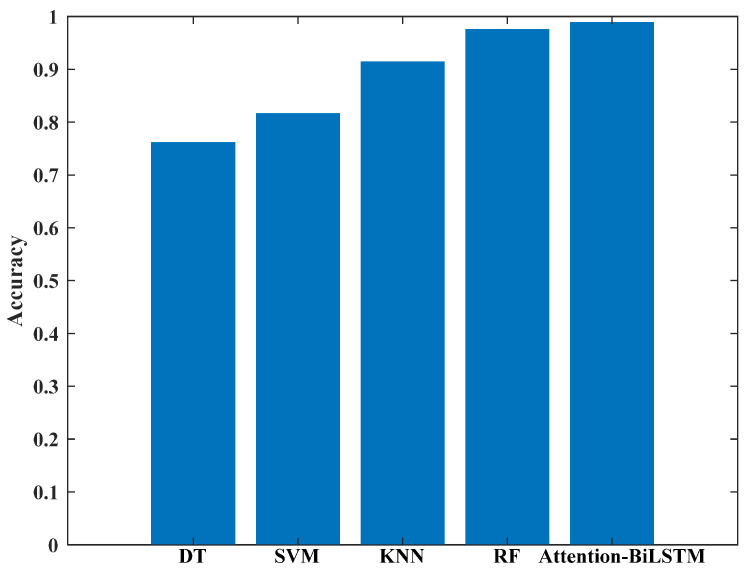
Average accuracy of the model.

**Figure 17 sensors-23-06693-f017:**
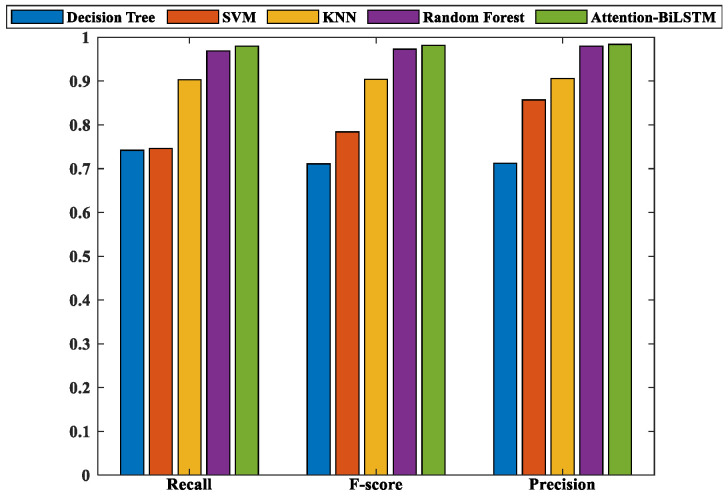
Related model evaluation index.

**Figure 18 sensors-23-06693-f018:**
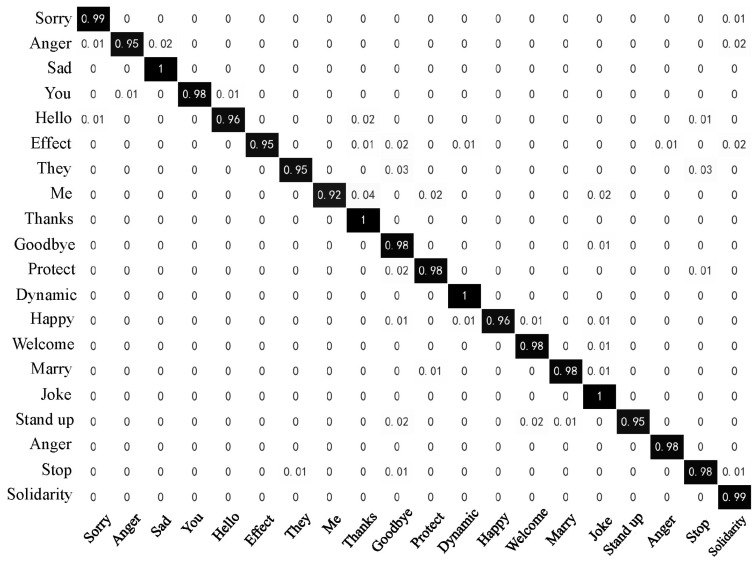
Confusion Matrix of Random Forest.

**Figure 19 sensors-23-06693-f019:**
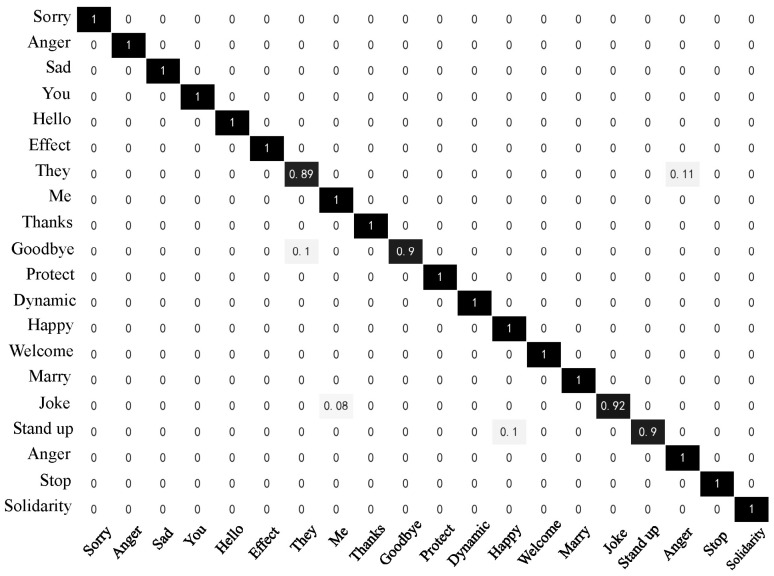
Confusion matrix of Attention-BiLSTM.

**Table 1 sensors-23-06693-t001:** mpu9250 specification.

Unit	Accelerometer	Gyroscope	Magnetometer
Dimensions (axis)	3	3	3
Dynamic Range	±16 g	±2000°/s	±4800 uT
Sensitivity	2048 LSB/g	16.4 LSB/(°/s)	0.6 uT/LSB
Bandwidth	4000 Hz	1000 Hz	8 Hz
Word Length	16 bits	16 bits	14 bits

**Table 2 sensors-23-06693-t002:** Sign language action details.

Sign Language	Description
Sorry	Five fingers together to make a “salute” gesture, stretch the little finger on the chest (one hand)
Anger	Hold one hand on the chest, then spread the palm upward (one hand)
Sad	Hold one hand to the chest and turn it a few times (one hand)
You	Point one index finger at each other (one hand)
Hello	Point one index finger at each other, then stick your thumb up (one hand)
Effort	Clench fist, bend elbow, swing left and right (one hand)
They	Point the index finger to the side, turn the palm down in front of the chest (one hand)
Me	Point to yourself with one hand (one hand)
Thanks	Stick out your thumb with one hand and bend it twice (one hand)
Goodbye	Raise one hand, stretch out five fingers naturally, and wave your wrist twice (one hand)
Protect	One hand sticks out the thumb, and the other hand wraps the thumb (both hands).
Dynamic	Hands clenched fist alternately in front of the chest, thumb, index finger into a “┖ ┚” shape, placed on both sides of the cheek, alternately up and down (hands)
Happy	Move your palms up and down on your chest (hands).
Welcome	Hands palms up, move aside, then clap hands (hands)
Marry	Stretch your thumbs with your fingertips opposite each other and bend twice (hands).
Joke	To wave (both hands) with one hand flat and the other in front of the palm.
Stand up	Stretch your hands flat, palms up, and move upward at the same time (hands)
Agree	To cross (hands) several times with fingers in front of the chest.
Stop	One hand is flat, the palm is down, and the other is against the palm (hands)
Solidarity	One palm up, the other palm down, hold each other and shake each other (hands)

**Table 3 sensors-23-06693-t003:** Summary of features extraction.

Feature	Description
Mean value	$\bar{X}=\frac{1}{n}\cdot \sum_{i=1}^{n}X_{i}$
Standard deviation	$X_{std}=\sqrt{{\frac{1}{n}\cdot \sum_{i=1}^{n}\left(x_{i}-\bar{X}\right)}^{2}}$
Skewness	$X_{sf}=\frac{\frac{1}{n}\cdot \sum_{i=1}^{n}{\left(\left|x_{i}\right|-\bar{X}\right)}^{3}}{{\sqrt{\frac{1}{n}\sum_{i=1}^{n}X_{i}^{2}}}^{3}}$
Kurtosis	$C_{k}=\frac{1}{n}\cdot \sum_{i=1}^{n}X_{i}^{4}$
Corrugation factor	$C_{s}=\frac{\sqrt{\frac{1}{n}\sum_{i=1}^{n}X_{i}^{2}}}{\frac{1}{n}\sum_{i=1}^{n}\left|X_{i}\right|}$
quartile	Upper quartile
Spectrum peak	The peak of Fourier change
The peak frequency of the spectrum	The frequency of the peak of Fourier change

**Table 4 sensors-23-06693-t004:** Nodes combination fetail.

Combination Method	Details
Combination 1	MAN1
Combination 2	MAN1, 2, 5, 8, 11, 14
Combination 3	MAN1, 3, 6, 9, 12, 15
Combination 4	MAN1, 4, 7, 10, 13, 16
Combination 5	Combination2 + Combination3
Combination 6	Combination2 + Combination4
Combination 7	Combination3 + Combination4
Combination 8	All MANs

**Table 5 sensors-23-06693-t005:** Summary of five-fold cross-validation accuracy.

Model	First Accuracy (%)	Second Accuracy (%)	Third Accuracy (%)	Fourth Accuracy (%)	Fifth Accuracy (%)	Average Accuracy (%)
DT	76.45	77.75	75.17	77.59	76.55	76.70
SVM	82.87	83.00	82.76	82.72	83.00	82.87
KNN	91.29	91.79	91.62	91.32	92.03	91.61
Random Forest	97.47	97.53	97.60	97.83	97.50	97.58
Attention-BiLSTM	99.25	99.25	98.50	98.75	98.50	98.85

**Table 6 sensors-23-06693-t006:** Statistics of model classification results.

Model	Accuracy (%)	Precision (%)	Recall (%)	F-Score (%)
DT	77.59	71.24	71.40	71.20
SVM	83.00	85.80	76.51	79.70
KNN	92.03	90.50	90.30	90.15
Random Forest	97.83	97.98	96.92	97.33
Attention-BiLSTM	98.19	98.39	98.03	98.15

**Table 7 sensors-23-06693-t007:** Accuracy comparison of classification models.

	Datasets	Waveglove-Single	Waveglove-Muti	Mhealth	Usc-had	Utd-Mhad1	Utd-Mhad2	Wharf	Wisdm	uWave
Model										
Baseline Decision Tree		99.10	96.63	93.41	89.22	67.51	81.90	66.26	61.61	70.72
DeepConvLSTM [24]		98.05	99.30	81.01	83.97	67.29	86.50	67.98	91.23	98.10
DCNN Ensemble [25]		-	-	93.09	88.49	62.03	81.63	75.50	89.01	-
Transformer-based [21]		99.40	99.99	90.35	89.83	76.32	88.42	78.63	84.53	98.80
Random Forest		96.22	99.53	98.50	94.44	68.55	87.67	77.32	96.80	86.92
Attention-BiLSTM		98.82	99.76	99.00	91.16	70.44	91.72	84.48	97.11	99.29

## Data Availability

The data presented in this study are available upon request from the corresponding author. The data are not publicly available due to the personal information of the data collectors.

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
