# Peer review of "Dataglove for Sign Language Recognition of People with Hearing and Speech Impairment via Wearable Inertial Sensors"

_sensors, 2023, doi:10.3390/s23156693_

Round 1

Reviewer 1 Report

Healing and speech impairment present significant challenges worldwide. This manuscript introduces a cost-effective data glove that utilizes multiple inertial sensors to achieve accurate and efficient sign language recognition. The authors demonstrate that the Attention-BiLSTM and RF algorithms outperform others in recognizing twenty dynamic sign language gestures. This research holds promise for fostering reliable communication between deaf and able-bodied individuals. However, to ensure a high-quality manuscript, it is essential to address several issues:

1.     The authors need to clarify the key challenge they addressed. The proposed method appears to have similarities with references 18-20, which also employed IMU and flexible capacitive/resistance sensors for sign language recognition. It is important to highlight the distinctive features and contributions of the proposed method.

2.     While the authors utilized four algorithm models, including SVM, DT, KNN, and RF, which are commonly used in deep learning, they should provide a more comprehensive explanation of the advancements and benefits of employing the attention-BiLSTM model.

3.     Regarding the higher recognition accuracy of combination 4 and the use of eight different combinations, the authors should elaborate on their reasoning. Is it because the distal knuckle allows for larger movements or higher acceleration? Furthermore, the rationale behind designing eight different combinations should be clearly stated.

4.     It is crucial to ensure a proper balance between the training and validation sets. The authors mentioned employing a five-fold cross-validation approach but did not provide details on the validation process. How many subsets were used for training, and what criteria were used for determining the adequacy of the training subsets?

By addressing these concerns, the manuscript will be enhanced, providing a clearer understanding of the research objectives, methodology, and results. This will contribute to the overall quality and impact of the study.

I humbly acknowledge that further improvements are needed in the language and formatting of the manuscript.

Reviewer 2 Report

The paper presents a very interesting topic concerning a low-cost data glove that utilizes multiple inertial sensors in order to recognize Chinese sign language. The paper is well-written; however it needs some improvements:

·       I suggest changing statement “This article centers on” to “This article is aimed at”;

·       I suggest changing “research” on “studies”;

·       In the introduction section the aim should be added;

·       I would delete the statement “Our research involves the following three main aspects:” in the introduction section;

·       I suggest adding information about number of data were collected, how the process of learning networks was performed. How were data divided into training, validation and testing?

·       Please verify statements “feature extraction” and “features extraction”;

·       I think in the statement “as shown in Table 1 The Data” the dot is missing ([age 3, line 148);

·       I suggest changing“subjects” to “participants”;

·       Fig. 6, 11, and 16 are illegible. I suggest increasing the font;

·       On page 7, line 237, the sentence should start with a capital letter;

·       On page 7, line 249 “where” should be written with lowercase. Before “And” there shouldn’t be a dot;

·        On page 7, the statement “Table 2 provides a description” should be changed to “Table 2 provides the description”;

·       After the statement “The following are the specific operations for data collection:” the listing of items should be added. Otherwise, the sentence should be ended.

·       In the statement “Figure 12 However” the dot is missing;

·       The statement should be corrected “Figure 18 and Figure 19 below display” – the figures are above.

Reviewer 3 Report

As the number of people with hearing impairments has increased, so the need for professional staff to help interpret deaf-mute sign language also increased. The problem is that the number of these people has not increased proportionally so the need to develop platforms capable of allowing problem-free communication between deaf-mute people or/and able-bodied has become quite important.

The authors of this paper present a glove equipped with inertial sensors that can be used for efficient and accurate sign language recognition.

To demonstrate the validity of the proposed solution, four machine learning models (decision tree, support vector machine, K-nearest neighbor, random forest) were used to recognize 20 types of dynamic sign language data. An attention-based neural network mechanism of long-term and short-term memory was also proposed, which proved to be quite efficient with a high detection rate, at the same level as the random forest learning model.

All this research would be pointless if the final product were not accessible to the masses, which first of all implies low costs but also increased autonomy. To this end, the authors evaluated the impact on the recognition and performance of the number and placement of the motion analysis nodes.

It is an interesting article, the data is presented clearly and the results obtained are easy to understand. The language used is very clear, there would be only one observation related to the bibliography. More precisely, certain references in the bibliography have some words divided into syllables, such as in reference 21 the word "multiple",  in reference 23 the word "personalized", etc.

Round 2

Reviewer 1 Report

The authors have addressed the main concerns. Now the manuscript can be published in the current version. 

Minor English polishment should be processed.